# Hsp70 Gene Family in *Sebastiscus marmoratus*: The Genome-Wide Identification and Transcriptome Analysis under Thermal Stress

**DOI:** 10.3390/genes14091779

**Published:** 2023-09-09

**Authors:** Xiaolu Han, Shihuai Jin, Chenyan Shou, Zhiqiang Han

**Affiliations:** Fishery College, Zhejiang Ocean University, Zhoushan 316002, China

**Keywords:** heat-shock proteins, marbled rockfish, heat stress, cold stress, phylogenetic analysis

## Abstract

Heat shock protein 70 kDa (Hsp70) is a highly conserved heat stress protein that is important in biotic processes and responses to abiotic stress. Hsp70 genes may be important in *Sebastiscus marmoratus,* for it is a kind of nearshore reef fish, and habitat temperature change is more drastic during development. However, genome-wide identification and expression analysis in the Hsp70 gene family of *S. marmoratus* are still lacking. Here, a total of 15 Hsp70 genes in the genome of *S. marmoratus* are identified, and their expression patterns were investigated using transcriptomic data from thermal stress experiments. The expansion and gene duplication events of Hsp70 genes from the Hspa4, Hspa8, and Hspa12a subfamilies in *S. marmoratus* are revealed by phylogenetic analysis. qRT-PCR expression patterns demonstrated that seven Hsp70 genes were significantly up-regulated and none were significantly down-regulated after heat treatment. Only the *hsp70* gene was significantly up-regulated after cold treatment. The selection test further showed a purifying selection on the duplicated gene pairs, suggesting that these genes underwent subfunctionalization. Our results add novel insight to aquaculture and biological research on *S. marmoratus*, providing important information on how Hsp70 genes are regulated in Scorpaeniformes under thermal stress.

## 1. Introduction

As a highly conserved protein, heat shock protein 70 kDa (Hsp70) could protect features in response to intense cellular stress by assisting in their refolding and stabilization [1,2]. By correcting protein folding, Hsps could minimize the aggregation of unnatural proteins and denatured proteins, which have a significant influence on resistance to adverse environmental stresses as a group of molecular chaperones [3,4,5,6]. Based on their function and weight, Hsps could be divided into several families, including Hsp110, Hsp90, Hsp70, Hsp60, and Hsp40, as well as some small heat shock proteins [7]. In those proteins, Hsp70 is one of the most extensively studied proteins because it is conserved in evolution. They widely exist in all organisms except some archaea [8,9,10].

*S. marmoratus*, commonly known as marbled rockfish, belongs to the family Scorpaenidae, order Scorpaeniformes, and is a kind of sedentary, warm-temperate, oviparous fish [11,12]. In the Western North Pacific, especially in China, Korea, Japan, and the Philippines, *S. marmoratus* is widely distributed and usually lives in the nearshore reef sea area without long migration [13,14]. Its meat is tender, has high nutritional value, and has potential value in aquaculture [15]. Due to its small size and ease of feeding, it has become the target of artificial breeding in China [16,17]. The suitable temperature of *S. marmoratus* is constantly changing during its growth period [18], ranging from 13 to 26 °C [19]. Additionally, the temperature of their habitat also varies dramatically. Consequently, strong adaptability to high and low temperatures means that the Hsp70 gene family may be very important in the temperature adaptation of *S. marmoratus*.

Global warming harms the growth and development of organisms [20]. Numerous studies have demonstrated that heat stress alters gene expression in plants and animals; for example, 14 StHsp20 genes in *Solanum tuberosum* were significantly up-regulated under heat stress [21]. The expansion or positive selection under thermal change of several *HSP* genes in crabs was detected based on full-length transcripts [22] and genome-wide analysis [23]. Therefore, studies on the identification and expression of Hsp gene families in organisms can provide insights into the protection of organisms under global warming.

The comprehensive description of the Hsp70 gene family has been studied in plants [24], fungi [25], insects [26], crustaceans [27], and mammals [28]. However, for the lack of fish genome resources and transcriptome sequences, genome-wide identification in fish has barely been carried out except for a few taxa, such as *Ictalurus punctatus*, *Larimichthys crocea*, and *Boleophthalmus pectinirostris* [8,29,30]. Specifically, 16 Hsp70 genes were identified in *I. punctatus*, showing significant up-regulation or down-regulation after bacterial infection [8]. In *L. crocea*, six Hsp70 genes out of 17 were significantly up-regulated or down-regulated under thermal stress [29]. Twenty Hsp70s were identified in *B. pectinirostris*, and eight genes were significantly involved in the high environmental ammonia (HEA) stress response [30]. However, genome-wide screening and transcriptome analysis of the Hsp70 gene family are still lacking in the Scorpaeniformes. We have completed the genome sequencing of *S. marmoratus*. These resources have made it possible to perform a comprehensive study of these gene families in the *S. marmoratus* genome.

Herein, genome-wide identification of Hsp70 genes in *S. marmoratus* was performed, and their gene expression levels under the stimulation of temperature were investigated. The findings will provide valuable resources for further studies on the evolutionary relationship of the Hsp70 gene in teleosts and the mechanism of Hsp70 gene regulation. The results could provide references to the molecular mechanism of Hsp70s in the adaptation of Scorpaeniformes under thermal stress and benefit offshore aquaculture for this high-economic species.

## 2. Results

### 2.1. The Hsp70 Genes in S. marmoratus Based on Genome Data Analysis

Twenty-seven candidate sequences were initially obtained from the HMM profiles of the HSP70 domain, *Hsp12a*, and *Hsp12b* (PF00012, PTHR14187: SF46, and PTHR14187: SF3). Some candidate sequences were deleted because of the absence of the Hsp70 domain. Finally, the presence of the HSP70 domain in 15 Hsp70 genes was confirmed, and we provided detailed information in the methods section. Fifteen Hsp70 gene members established robust nomenclature in the Zebrafish Nomenclature Guidelines. The CDS of these Hsp70 genes varied from 1320 to 3054 bp in length and encoded proteins from 439 to 1017 amino acids (aa). The predicted physicochemical properties of the amino acid sequences showed that *hspa13* had the shortest conserved domain of 389 aa, and *hspa4l* had the longest conserved domain of 688 aa. The Hsp70 proteins had molecular weights of 48108.26–113767.84 kDa. Their predicted isoelectric points (pI) ranged from 4.98 (*hspa5*) to 8.73 (*hspa12a*). Fifteen Hsp70 genes were spread over eleven chromosomes. As shown in Table 1, *hspa4*, *hyou1*, and *hspa8a* were distributed on the same chromosome.

### 2.2. The Phylogeny of the Hsp70 Genes among Species

We constructed an unrooted phylogenetic tree of the Hsp70s from the amino acid sequence of *S. marmoratus* and the classic model species to analyze the evolutionary relationship and classification of this gene family (Figure 1). Two human-related genes, HSPA8 (named *hspa8a* and *hspa8b* according to the cluster), were detected in the genomes of six teleost fish. The names of the Hsp70 genes in *S. marmoratus* were standardized following the zebrafish and human orthologs. From the phylogenetic relationships, all the Hsp70 gene family members in *S. marmoratus* were located in different branches and clustered with the corresponding genes from other fish species at first with good bootstrap values, indicating that Hsp70 genes corresponding to different species are more closely related than the different Hsp70 genes of the same species.

We identified 12 homologous Hsp70 genes (present similarly in humans) in *S. marmoratus*, but *hspa2*, *hspa6,* and *hspa7* were not identified. The *hsph1* gene didn’t appear in any other teleosts except zebrafish. Most orthologous pairs among *S. marmoratus* and the other five fish species have been confirmed, which suggests that the same ancestor gene of the Hsp70 family may have appeared before the genetic differentiation of fish. Furthermore, three fish genes (*hspa8a*, *hspa8b*, and *hsc70*) might be closely related paralogs of the human HSPA8 gene (*hsc70* vs. *hspa8a* and *hspa8b*).

### 2.3. Gene Structure, Motif, and Chromosomal Location Analysis

The intron–exon structure diagram of every Hsp70 gene in *S. marmoratus* is shown in Figure 2. We performed the structure analysis based on a genome annotation file. Introns existed in all genes, ranging from 1 to 23. Among them, two members (*hspa1b*, *hsp70*) just had one exon; however, the number of exons in the remaining genes ranged from 4 to 23. The number of introns and exons varies greatly among different genes, indicating that these genes may have some differences in biological functions.

Conserved motif analysis was performed on the Hsp70s of *S. marmoratus* to understand its functional diversity. We searched for each Hsp70s with fifteen putative motifs (Figure 3). In addition, there was only one motif in *hspa12a* and *hspa12b*; motif 15 is present only in *hspa4a*, *hspa4b*, *hspa4l,* and *hspa14*; and motifs 1, 2, 3, 4, and 8 were present in the remaining 13 genes. Among them, some genes had similar or even common motif compositions that were closely related to the phylogenetic relationship, and they were often found in close evolutionary clusters. To some extent, the motif analysis results could support the phylogenetic relationship of the Hsp70 gene family.

The identified Hsp70 genes were sketched on chromosomes to investigate the *hsp70* expansion events in *S. marmoratus*. As shown in Figure 4, 15 Hsp70 genes were distributed on 11 chromosomes. Of all Hsp70 genes, three were located on chromosome 16, which had the largest number of Hsp70 genes; two members were distributed on chromosomes 8 and 10, respectively; and the remaining chromosomes contained only one member. Our results suggest that the *hsp70* expansion may occur after two rounds of whole genome duplication (WGD) in early vertebrate evolution, for the *hspa12a* and *hspa12b* (which existed in most vertebrates) are identified at different chromosomes [31].

As shown in Figure 5, the Hsp70s in *S. marmoratus* are highly conserved and further verified using the CDD and Pfam databases. All Hsp70 proteins contain a highly conserved domain between 3 and 765 aa at the N terminal, the presence of which confirms that they are Hsp70 proteins.

### 2.4. Protein Signal Peptides Predictive Analysis

Our analysis suggested that Hspa5 and Hyou1 were identified as secreted proteins by signal peptide prediction analysis. The highest values of the raw cleavage site score (CS) of two predicted proteins were 0.9823 and 0.5999, which appeared on the 16th alanine (A) and 26th valine (V), respectively. Our results suggested that Hyou1 and Hspa5 had signal peptides based on the SP value, and their lengths were approximately 15 and 25 amino acids, respectively. The results of the protein signal peptide predictive analysis can be found in Appendix A.

### 2.5. Secondary and Third Structure Prediction, Subcellular Localization of Hsp70 Proteins

Random coils and α helices were the main secondary structures of the 15 proteins encoded by the *S. marmoratus* Hsp70 genes. α helices accounted for 30.89% (Hspa12a)–47.89% (Hyou1), β turn accounted for 3.11% (Hspa4l)–7.74% (Hspa13), random coil accounted for 29.71% (Hspa9)–44.66% (Hspa12a), and extended strand accounted for 12.39% (Hyou1)–22.13% (Hspa14) (Table 2, Figure 6).

All Hsp70 gene members of *S. marmoratus* were expressed in the mitochondrion, plasma membrane, cytoplasm, endoplasmic reticulum, and nucleus. Hspa5 and Hyou1 proteins were expressed in the endoplasmic reticulum. Hspa9 and Hspa8a proteins were expressed in the mitochondrion. The Hspa12a protein was expressed in the nucleus. The rest of the proteins were all expressed in the cytoplasm (Table 2).

### 2.6. Selection Test on Duplicated Hsp70 Gene Pairs

By calculating the Ka, Ks, and Ka/Ks for five homologous Hsp70 gene pairs, we assess the evolutionary constraints and selection pressures on the Hsp70 genes. The synonymous substitution rate (Ks) represents the background base substitution rate, and we could use Ks values to evaluate the WGD times. Our results suggest that strong purifying selection may have occurred during evolution because the Ka/Ks values of the Hsp70 gene pairs were all less than 1.0 (Table 3).

### 2.7. Expression Profiles of Hsp70 Genes under Thermal Stress Treatment

To understand the gene expression profiles of Hsp70s in *S. marmoratus*, high-temperature and low-temperature stress experiments were conducted, and corresponding transcriptome analysis was performed on its liver. Ten Hsp70 genes were expressed in liver tissue among the 15 Hsp70 genes (Table 4; Figure 7), which indicated a tissue expression pattern. Under high-temperature treatment, seven (*hspa8a*, *hspa8b*, *hspa9*, *hsc70*, *hyou1*, *hspa4a*, and *hspa5*) Hsp70 genes were significantly up-regulated (log2FC: 1.32–6.54), while no significantly down-regulated genes were found. Under low-temperature stress treatment at 13 °C, only *hsp70* was significantly up-regulated. The expression levels of the remaining 4 Hsp70 genes (*hspa4b*, *hspa5*, *hspa13*, and *hspa14*) exhibited little change under cold or heat stress.

### 2.8. Validation of Transcriptomic Data via qRT-PCR

Five significant expressed genes in the high-temperature group were used for qRT-PCR analysis to verify the accuracy of all the transcriptome data (Figure 8). The qRT-PCR results showed a similar tendency to the RNA-Seq results, indicating credible transcriptome data despite slight value discrepancies.

## 3. Discussion

Hsp70s are essential in responding to biotic and environmental stress. In recent years, several studies have been conducted on the Hsp70s of fish; however, genome-wide identification and expression analysis have not yet been conducted in Scorpaeniformes. In the current study, the whole Hsp70 gene family of *S. marmoratus* under thermal stress was well investigated, including phylogeny, functional characterization, selection tests, and expression pattern analysis in the liver. These results provide a new way to study this physiological response and the regulatory mechanism of *S. marmoratus* against thermal stress and are a valuable theoretical basis for ovoviviparous fish aquaculture.

The literature often uses the member name of this gene family in confusion, resulting in a series of naming-related problems. Therefore, when the term Hsp70 is used, it is difficult to know which gene or protein in the family is described without further description. A total of 15 Hsp70 genes were identified and annotated in *S. marmoratus* and based on the evolutionary relationship, the names of Hsp70 proteins were well assigned (the identified Hsp70 genes can be found in Appendix A). We searched exhaustively in the genomic resources, but the *hsph1* gene was still not found in *S. marmoratus*. It is unclear whether *hsph1* is truly missing or whether it is due to incomplete sequencing or annotation of the *S. marmoratus* genome. The *hsph1* gene was only found in *Danio rerio* and *L. crocea* and was absent in other fish species, including *S. marmoratus*, *Oncorhynchus mykiss*, *B. pectinirostris*, *O. niloticus*, and *Takifugu rubripes*. In six fish, the *hspa6* and *hspa7* genes were not found in any species. Compared to humans, most Hsp70 genes were found in *S. marmoratus*, except *hspa2*, *hspa6*, *hspa7*, and *hsph1*. Compared to *L. crocea*, although *S. marmoratus* had fewer copies of the *hspa4l* and *hspa5* genes, most Hsp70 family genes of *S. marmoratus* and *L. crocea* were clustered together. They had high homology, as revealed by the phylogenetic tree. When compared to *B. pectinirostris*, the *hspa4l*, *hspa8b*, and *hsc70* genes of *S. marmoratus* seemed to have been lost in *B. pectinirostris* [30]. Furthermore, *B. pectinirostris* has two more copies of *hspa8a* than *S. marmoratus* (*hspa8a.1*, *hspa8a.2*) [30], and the same result can also be found in *I. punctatus* [8]. The present study suggested that no tandem duplication of *hsp70* was observed in *S. marmoratus*, whereas previous studies have shown that the tandem duplications of *hsp70* (named *hspa1a* in *B. pectinirostris*) paralogs have been described in *B. pectinirostris*, *I. punctatus,* and *D. rerio* [8]. Since the number of tandem duplication genes differs among species, it may be closely related to the species adaptation to the environment.

Differences in the biological processes of subcellular compartments may be relevant to distinct functions in the Hsp70 gene structure [32]. Subcellular localization has shown that Hsp70 family members are located in the cytoplasm, mitochondria, plasma membrane, endoplasmic reticulum, and nucleus [33]. It has been demonstrated in past research that cytoplasmic Hsp70s play an important role under stressed and even non-stressed conditions [34,35]. Analysis of signal peptide prediction has shown that Hyou1 and Hspa5 were preliminary determined as secreted proteins. According to the chromosome location analysis, the Hsp70 gene members were located on different chromosomes. It is worth mentioning that when gene pairs are located on different chromosomes, the gene replication process can be considered fragmented [36]. Instead, tandem duplications refer to the copies of genes on the same chromosome [37]. Duplicated *hsp70* genes (*hspa4a*, *hspa4b*, *hspa4l*, *hspa8a*, *hspa8b*, *hspa12a*, and *hspa12b*) were also found in *S. marmoratus*, analogous to their function by their gene structure and motifs.

Gene duplications occur in all kinds of life forms and provide the raw material for functional innovation [38]. Gene replication is the main mechanism for gene family amplification through WGD or single-gene duplication (SSD) in the process of evolution [39]. The phylogenetic tree in this study found that mammals tend to have fewer duplications than teleosts, which may be related to the teleost undergoing whole genome duplication. Compared with tetrapods (amphibians, reptiles, birds, and mammals), more paralogues of *hspa8* and *hspa4* were found in teleosts (referring to *hspa8a* and *hspa8b*, *hspa4a* and *hspa4b*, respectively). The teleost-specific genes indicated that *hspa8a* and *hspa8b*, *hspa4a* and *hspa4b* were present after the divergence between teleosts and tetrapods, which is consistent with the research results of Xu et al., 2018 [29].

According to the thermal stress experiment of *S. marmoratus*, 10 out of 15 Hsp70 genes were expressed in liver tissue. Based on these results, it can be inferred that they are involved in thermal adaptation and have different expression patterns under heat stress, which may be related to the different effects of cold and heat on physiology. Seven genes (*hspa4a*, *hspa5*, *hyou1*, hspa8a, *hspa8b*, *hspa9,* and *hsp70*) were significantly up-regulated and no significantly down-regulated genes were found under high-temperature treatment. In terms of high-temperature treatment, our results were the same as those from the temperature trials of *L. crocea* [29]. Severe cell damage can be caused by extreme temperatures, and different species of fish involve the rapid synthesis of Hsp70, which aims for cellular protection by stabilizing protein acquisition resistance and thermo-tolerance [40,41]. Only one gene (*hsp70*) was significantly up-regulated, and other genes were slightly changed under low-temperature treatment. In general, heat and cold stress are opposite to the regulation of enzyme reactions, diffusion, and membrane transport rates [42]. Under low-temperature treatment, one significantly up-regulated gene (*hsp70*) was found. Living organisms utilize various mechanisms to manage and repair damage caused by different types of cellular stressors [43]. So, at 13 °C, the *hsp70* gene was highly expressed, protecting cells from the effects of low-temperature stress, including protein damage and misfolding. The number of damaged proteins increased under low-temperature stress (13 °C). The production of *hsp70* gradually increased, presumably reflecting the need for more Hsp70 protein, which promoted the renaturation of abnormal proteins [1]. The *hyou1* and *hspa5* proteins have a signal peptide and a transmembrane domain and are found in the endoplasmic reticulum as secreted proteins. The *hyou1* and *hspa5* proteins, as companion proteins of the endoplasmic reticulum (ER), play a crucial role in protein folding and ensuring the quality control of protein synthesis in ER tubes. As the temperature decreases, protein synthesis in the ER is inhibited. Thus, the expression of the *hspa5* and *hyou1* genes is involved in the correct folding of proteins, and the degradation of misfolded proteins is limited by low-temperature stress. Expression of *hspa5* and *hyou1* under heat stress is temperature-induced. Similar results were found in *O. mykiss* [44].

## 4. Materials and Methods

### 4.1. Animal Materials and Experiment

In order to obtain the expression profile of Hsp70 genes under thermal stress, temperature stress experiments were conducted, and the RNA-seq reads were subsequently obtained. Briefly, several *S. marmoratus* individuals with similar morphology and physiology were collected in July 2019 and temporarily cultivated in the laboratory. The water temperature in the plastic pool was kept at 20 °C and the salinity was kept at 20‰. To reduce the unquantifiable influences on transcriptome sequencing, feeding was stopped and oxygen was continuously charged during the temporary-keeping period; the dissolved oxygen value remained at about 8mg/L. The suitable temperature of *S. marmoratus* is constantly changing during its growth period, ranging from 13 to 26 °C [18,19]. Three temperature groups, with ten fish in each group, were assigned as the low-temperature group (13 °C), the control group (20 °C), and the high-temperature group (25 °C), respectively. The trial lasted 6 h. Transcriptome sequencing was performed on the liver tissues of three fish randomly selected from each group. All fish were anesthetized with eugenol (Eugenol [CAS 97-53-0], Cat. No. CN00359) for dissection. These specimens were preserved in the RNA preservation solution and stored at −80 °C for later use. The total RNA was extracted using the TRIzol Reagent Kit (Invitrogen, Carlsbad, CA, USA). The RNA integrity number (RIN) of the RNA samples was 6.7, and the total RNA concentration was more than 195ng/ul. We used the NEBNext Ultra RNA Library Prep Kit for Illumina1 (NEB, Ipswich, MA, USA) to construct a sequencing library. The sequencing platform used was the Illumina Hiseq 2500. All of the above is completed in accordance with the manufacturer’s instructions.

### 4.2. Identification of Hsp70 Members

The genome and protein sequence of *S. marmoratus* were completed by the fishery ecology and biodiversity laboratory of Zhejiang Ocean University (https://doi.org/10.6084/m9.figshare.19161290) (accessed on 22 July 2023). The hidden Markov model (HMM) profile of the Hsp70s gene was obtained from the Pfam protein families database (http://pfam.Xfam.org/) (accessed on 10 August 2023) [45], and the HMM profiles of *Hsp12a* and *Hsp12b* (PTHR14187: SF46 and PTHR14187: SF39) were downloaded from the PANTHER classification system (http://www.pantherdb.org/) (accessed on 10 August 2023). HMMER 3.2.1 software was used to identify Hsp70 genes from the *S. marmoratus* genome [46]. The online program Pfam and the NCBI conserved domain database (http://www.ncbi.nlm.nih.gov/cdd/) (accessed on 11 August 2023) were used to survey the conserved domains of the candidate protein sequence [47]. Furthermore, the fully conserved domain sequences from the *S. marmoratus* genome were retained as queries to search against this species in RNA-seq datasets. The online software SOPMA was utilized to predict protein secondary structure (http://npsa-pbil.ibcp.fr/cgi-bin/npsa_automat.pl?page=npsa_sopma.html) (accessed on 11 August 2023) [48]. WoLF PSORT (https://wolfpsort.hgc.jp/) (accessed on 11 August 2023) software was used to predict subcellular localization by uploading protein sequences [49]. The ExPASy ProtParam tool (https://web.expasy.org/protparam/) (accessed on 11 August 2023) was used to calculate protein physicochemical parameters such as molecular weight (MW) and isoelectric point (pI) [50].

### 4.3. Phylogenetic Relationship Analysis

We used the protein sequences identified from *S. marmoratus* and 13 other species (including humans, fish, amphibians, and crabs) to construct phylogenetic trees. The protein sequences were retrieved from the NCBI (http://www.ncbi.nlm.nih.gov) (accessed on 22 July 2023), UniProt (http://www.uniprot.org) (accessed on 22 July 2023), and Ensemble (http://asia.ensembl.org/index.html) (accessed on 22 July 2023) databases (detailed information can be found in Appendix A), as listed by Song et al., 2016 [8]. The *Portunus trituberculatus* sequences were downloaded from Jin et al., 2022 [27]. The ML analysis was employed to conduct phylogenetic and evolutionary analyses utilizing RAxML based on a PROTGAMMAIJTT matrix-based model with 1000 bootstrap replicates [51]. Muscle software was used to align multiple protein sequences.

### 4.4. Gene Structure, Motif, Chromosomal Location, and Conserved Domain Analysis

The structure, motif, and chromosome position of *S. marmoratus* Hsp70 genes were analyzed and visualized using TBtools v1.09861 [52] based on the genome annotation files. Conserved amino acid sequences of Hsp70 proteins in *S. marmoratus* were detected by MEME (Multiple Expectation Maximization for Motif Elicitation, version 5.4.1) [53] with the following parameters: site distribution was set at 0 or 1 occurrence per sequence, the maximum number of motifs to be found was 15, and the motif width was set to 10~50. The protein structure was visualized using DOG 2.0 with default parameters [54].

### 4.5. Three-Dimensional Structure Analysis of Hsp70 Proteins

Three-dimensional protein structures for various Hsp70 proteins were constructed through homology modeling using Swiss Model software (https://swissmodel.expasy.org/) (accessed on 12 August 2023) with default parameters, and the quality of the resulting models was assessed using SAVES v.6.0 (https://saves.mbi.ucla.edu/) (accessed on 12 August 2023) [55,56]. When the sequence identity of the module is less than 30%, then use Phyre2 online software (http://www.sbg.bio.ic.ac.uk/phyre2/html/page.cgi?id=index) (accessed on 12 August 2023) to generate 3D structure models with the highest confidence [57].

### 4.6. Predictive Analysis of Protein Signal Peptides and Selection Test of Hsp70s

Signal-peptide prediction analysis used the SignalP 6.0 online tool [58]. For the selection-pressure test, Hsp70 duplicated gene pairs from *S. marmoratus* were investigated through phylogenetic relationships. KaKs_Calculator2.0 software was used to calculate the rates of synonymous (Ks) and nonsynonymous (Ka) substitutions and their ratios [59].

### 4.7. Expression Pattern Analysis of Hsp70 Genes in S. marmoratus

The index of the transcript to the coding sequence map was created using Bowtie2 v2.5.0. Differential transcript expression levels were evaluated using RNA-Seq by Expectation Maximization (RSEM) [60]. We used the rsem-calculate-expression Perl script in RSEM 1.3.1 to quantify expression levels through Bowtie2 alignment of reads. The expression estimates of Hsp70 genes were normalized in the form of fragments per kilobase of transcript sequence per million fragments mapped (FPKM). The FPKM values were ≥5. Then, log2-based fold change (log2FC) values were calculated. Genes with *t*-test values (*p* < 0.05) and log2FC < −1.0 or log2FC > 1.0 were defined as differentially expressed genes.

### 4.8. Primer Design, RNA Isolation, and Quantitative Real-Time PCR (qRT-PCR) Validation

The total RNA of the samples was extracted using a standard Trizol Reagent Kit following the manufacturer’s protocol. RNA purity was checked using the NanoPhotometer^®^ spectrophotometer (IMPLEN, Westlake Village, CA, USA). RNA integrity was assessed using the RNA Nano 6000 Assay Kit of the Agilent Bioanalyzer 2100 system (Agilent Technologies, Santa Clara, CA, USA). Quantitative reverse transcription PCR (qRT-PCR) was performed to validate the transcriptomic data. Based on the DEG analysis, five genes with significant expression were used in the qRT-PCR analysis, and primers were designed using Primer Premier 5.0 (Appendix A). In addition, β-actin was selected as a reference gene for internal standardization. qRT-PCR was designed following the manufacturer’s instructions for TB Green^®^ Premix Ex TaqTM (Tli RNase H Plus) RR420A. Using the ABI PRISM 7300 real-time PCR system (APPlied Biosystems, Carlsbad, CA, USA), the reactions were prepared in a total volume of 20 μL, containing 2 μL of diluted cDNA template, 0.4 μL of each primer (10 μM), 10 μL of TB Green Premix Ex Taq (Tli RNase H Plus) (2×), 6.8 μL of nuclease-free water and 0.4μL ROX Reference Dye (50×). The PCR program was 95 °C for 30 s, followed by 40 cycles of 95 °C for 5 s and 60 °C for 30 s. Three parallel experiments on each cDNA template were performed to strengthen the veracity of the result. Relative expression levels of genes were calculated using the 2-ΔΔCT (ΔCT = CTtarget gene − CTreference gene, ΔΔCT = ΔCTtreatment − ΔCTcontrol) method.

## 5. Conclusions

This study represented a comprehensive genomic and transcriptomic analysis of Hsp70 genes in *S. marmoratus*. Fifteen Hsp70 genes were identified for the first time in *S. marmoratus*. All members of the Hsp70 gene family were well classified, and the expansion of Hsp70 genes from the Hspa4, Hspa8, and *Hspa12a* subfamilies in *S. marmoratus*. Regulated expression of Hsp70s was observed; some inducible Hsp70s were significantly up-regulated after thermal stress exposure, and they were from the expanded Hspa4 and Hspa8 subfamilies. These findings provide a reference for further research on the biology of *S. marmoratus* and help to illuminate the regulatory mechanism of the Hsp70 gene family in response to thermal stress. Above all, our results promoted awareness of the temperature adaptation mechanism of marbled rockfish and presented valuable evidence for this species’ aquaculture. Nevertheless, the lack of knowledge of physiological biochemistry, further verifications are needed to obtain more intuitive and accurate results to predict the exact implications of these changes.

## Figures and Tables

**Figure 1 genes-14-01779-f001:**
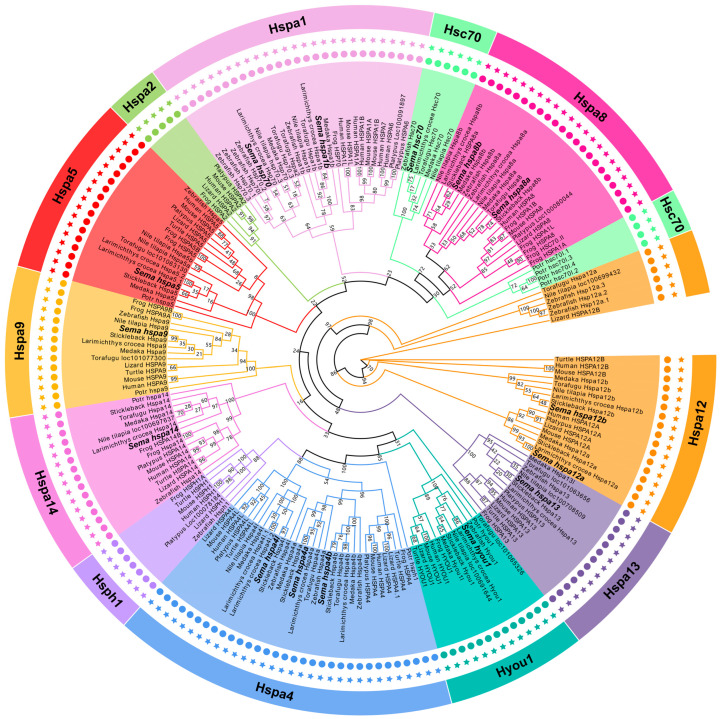
The phylogenetic tree of Hsp70 protein sequences among all species. The multiple sequence alignment was conducted by Muscle, and RAxML (raxmlHPC-HYBRID-SSE3) was employed for maximum likelihood (ML) analysis with 1000 bootstrap replicates. Hsp70 genes from *S. marmoratus* are in bold.

**Figure 2 genes-14-01779-f002:**
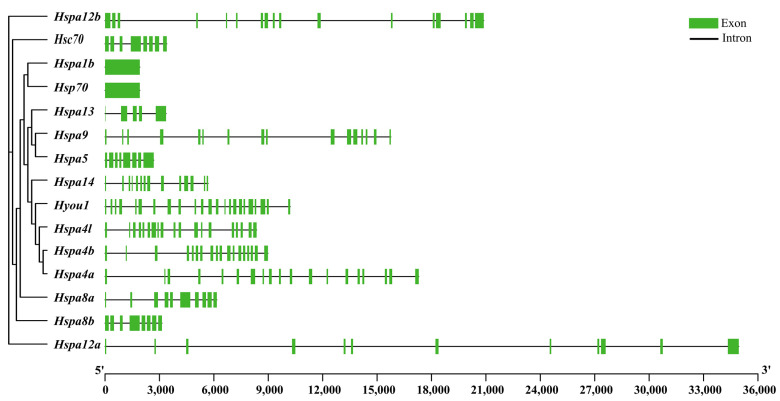
Hsp70 gene family member structure analysis of *S. marmoratus*. Green boxes represent exons, and the black line represents introns. Visualized by TBtools.

**Figure 3 genes-14-01779-f003:**
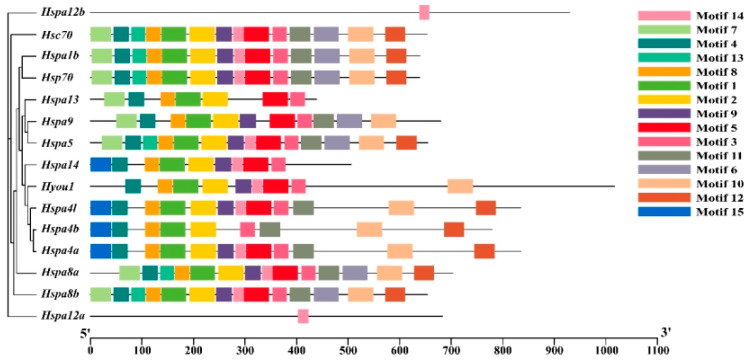
Distribution of conserved motifs of Hsp70 proteins from *S. marmoratus*. Motif visualized by TBtools.

**Figure 4 genes-14-01779-f004:**
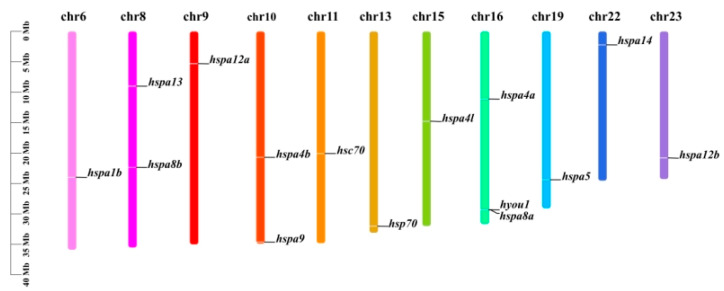
The location of Hsp70 genes in the chromosomes of *S. marmoratus*.

**Figure 5 genes-14-01779-f005:**
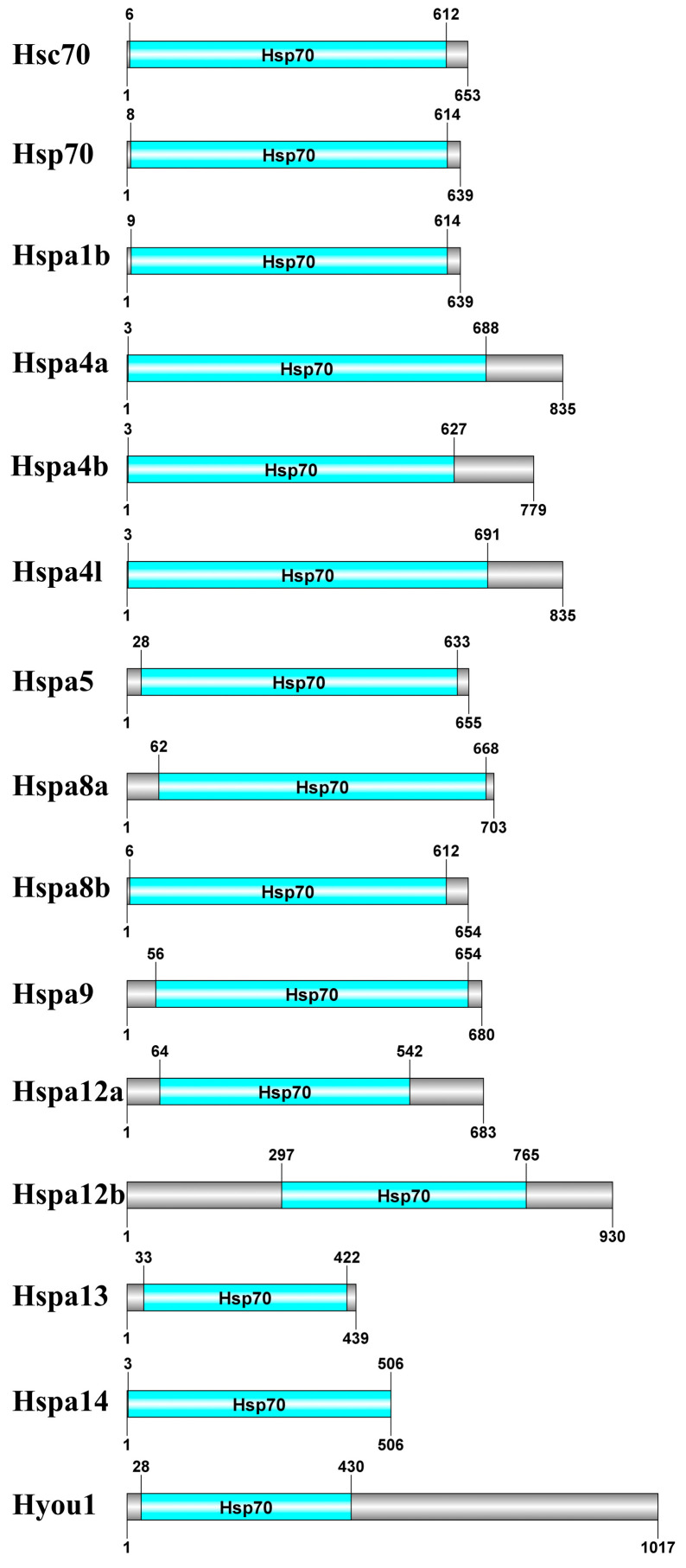
Conserved domain predictions for 15 Hsp70 proteins. Blue boxes and gray bars represent the conserved domains and the length of each protein sequence, respectively.

**Figure 6 genes-14-01779-f006:**
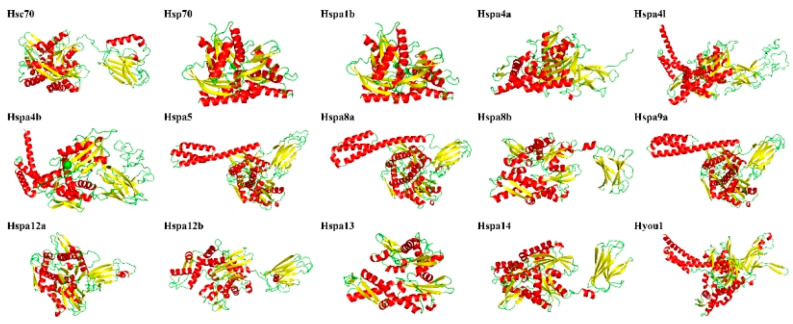
The 3-dimensional structures of Hsp70 proteins. The secondary structure elements include α helix (red), β turn (yellow), and random coil (green).

**Figure 7 genes-14-01779-f007:**
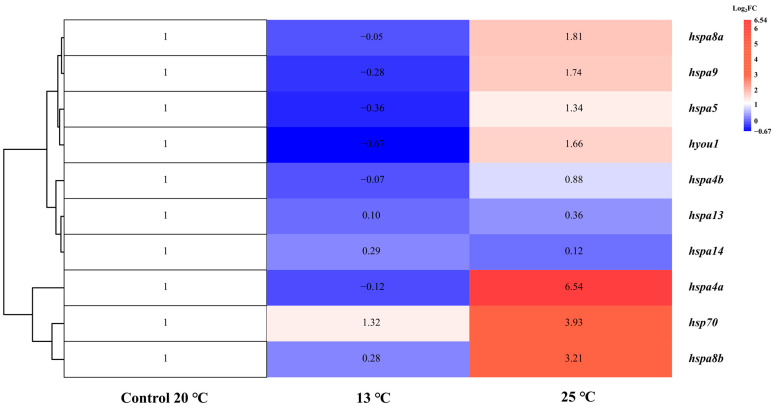
Expression profiles of *S. marmoratus* Hsp70 gene family in the liver. Heat map of *hsp70* expression under thermal stress treatments based on the fold change (log2) in FPKM values.

**Figure 8 genes-14-01779-f008:**
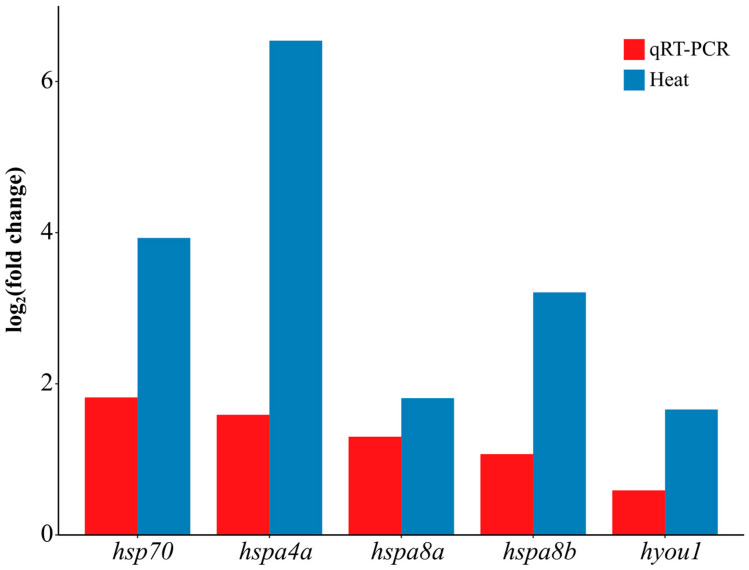
Validation of five significantly differentially expressed genes in the high-temperature group by Quantitative Reverse Transcription PCR (qRT-PCR).

**Table 1 genes-14-01779-t001:** Summary of 15 Hsp70 gene family members identified in *S. marmoratus*.

No.	Gene Name	Gene ID	CDS * Length (bp)	Protein Length (aa)	Hsp70 Domain Location (aa)	Molecular Weight (kDa)	Theoretic al pI *	Chromosome	Location
1	*hsc70*	Seb016770	1962	653	6-612	71,379.77	5.19	chr11	20077654:20081059
2	*hsp70*	Seb019324	1920	639	8-614	70,157.35	5.41	chr13	32013287:32015206
3	*hspa1b*	Seb001220	1920	639	9-614	70,308.53	5.44	chr6	23970762:23972681
4	*hspa4a*	Seb020794	2508	835	3-688	93,851.78	5.61	chr16	11119016:11136314
5	*hspa4b*	Seb006269	2340	779	3-627	87,193.47	5.06	chr10	20703166:20712148
6	*hspa4l*	Seb001425	2508	835	3-691	93,694.10	5.50	chr15	14780364:14788721
7	*hspa5*	Seb006799	1968	655	28-633	72,285.76	4.98	chr19	24390720:24393407
8	*hspa8a*	Seb009609	2112	703	62-668	76,830.31	5.47	chr16	29350778:29356943
9	*hspa8b*	Seb013994	1965	654	6-612	71,622.94	5.24	chr8	22354391:22357531
10	*hspa9*	Seb014614	2043	680	56-654	73,556.67	6.24	chr10	34615309:34631069
11	*hspa12a*	Seb017726	2052	683	64-542	76,345.16	8.73	chr9	5309800:5344743
12	*hspa12b*	Seb009063	2793	930	297-765	103,325.02	6.86	chr23	20798860:20819742
13	*hspa13*	Seb000354	1320	439	33-422	48,108.26	5.35	chr8	9021496:9024862
14	*hspa14*	Seb019773	1521	506	3-506	54,473.08	5.81	chr22	2233197:2238880
15	*hyou1*	Seb009605	3054	1017	28-430	113,767.84	5.38	chr16	29310839:29321048

* CDS: coding sequence. pl: isoelectric points.

**Table 2 genes-14-01779-t002:** Secondary structure prediction and subcellular location prediction of Hsp70 proteins in *S. marmoratus*.

Protein	α Helix	β Turn	Random Coil	Extended Strand	Subcellular Location Prediction
Hsc70	41.65%	7.04%	33.38%	17.92%	Cytoplasm
Hsp70	41.94%	7.04%	32.71%	18.31%	Cytoplasm
Hspa1b	42.41%	6.73%	31.77%	19.09%	Cytoplasm
Hspa4a	43.47%	3.47%	38.32%	14.73%	Cytoplasm
Hspa4b	40.56%	3.34%	40.95%	15.15%	Cytoplasm
Hspa4l	43.11%	3.11%	40.00%	13.77%	Cytoplasm
Hspa5	43.66%	7.18%	30.23%	18.93%	Endoplasmic reticulum
Hspa8a	39.40%	7.68%	35.14%	17.78%	Mitochondrion
Hspa8b	42.66%	7.03%	32.42%	17.89%	Cytoplasm
Hspa9	43.09%	7.35%	29.71%	19.85%	Mitochondrion
Hspa12a	30.89%	4.39%	44.66%	20.06%	Nucleus
Hspa12b	32.90%	5.91%	41.72%	19.46%	Cytoplasm
Hspa13	40.32%	7.74%	31.66%	20.27%	Plasma membrane
Hspa14	37.15%	5.73%	34.98%	22.13%	Cytoplasm
Hyou1	47.89%	3.44%	36.28%	12.39%	Endoplasmic reticulum

**Table 3 genes-14-01779-t003:** Ka/Ks * values of homologous Hsp70 gene pairs.

Gene Pair	Ka	Ks	Ka/Ks
*hspa12a-hspa12b*	0.2870	3.9431	0.0728
*hspa4a-hspa4b*	0.1788	4.2046	0.0425
*hspa4a-hspa4l*	0.2649	3.9346	0.0673
*hspa4b-hspa4l*	0.2861	3.7453	0.0764
*hspa8a-hspa8b*	0.0322	1.8155	0.0177

* Ka: nonsynonymous substitution rate; Ks: synonymous substitution rate.

**Table 4 genes-14-01779-t004:** Log2-based fold change (log2FC) of *S. marmoratus* Hsp70 gene expression in liver under thermal stress. The significant genes (*p* value < 0.05, FPKM number > 5, absolute log2FC > 1.0) and their fold changes are in bold.

Gene Name	log_2_FC		
	Control Group	Low-Temperature	High-Temperature
*hsp70*	1	**1.32**	**3.93**
*hspa4a*	1	−0.12	**6.54**
*hspa4b*	1	−0.07	0.88
*hspa5*	1	−0.36	**1.34**
*hspa8a*	1	−0.05	**1.81**
*hspa8b*	1	0.28	**3.21**
*hspa9*	1	−0.28	**1.74**
*hspa13*	1	0.10	0.36
*hspa14*	1	0.29	0.12
*hyou1*	1	−0.67	**1.66**

## Data Availability

The raw sequence data reported in this paper have been deposited in the Genome Sequence Archive in the National Genomics Data Center, China National Center for Bioinformation/Beijing Institute of Genomics, Chinese Academy of Sciences (GSA: CRA012447) that is publicly accessible at https://ngdc.cncb.ac.cn/gsa; The fifteen identified Hsp70 genes of *S. marmoratus* have been uploaded as fasta file in Appendix A.

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
