# Peer review of "Hsp70 Gene Family in Sebastiscus marmoratus: The Genome-Wide Identification and Transcriptome Analysis under Thermal Stress"

_genes, 2023, doi:10.3390/genes14091779_

Round 1

Reviewer 1 Report

Dear authors,

Thank you for the very interesting and rewarding manuscript. I am honored to be in charge of this review.

According to my review, some issues need to be clarified concerning the experimental design and methodology.

1. What is the scientific basis on which temperatures were chosen? Reference is required.

2. Line 307, oxygen was continuously charged. Dissolved oxygen value should be provided.

3. Why did the authors choose liver tissue? Is there any relation between it and the Hsp70 member’s expression?

4.  It is not mentioned the method used to euthanize the fish. Ethics rules are mandatory.

5. Keywords should be changed to be different from the title words

6. The introduction is too short. Need to add some effects of different temperatures on gene expression and should be ended with experiment aims.

7. Moderate English changes are required in all manuscript

Moderate English changes are required in all manuscript

Author Response

Dear reviewer,

Thank you for your valuable comments and for giving us the opportunity to revise the manuscript. We have carefully read your review comments and respond to your concerns point by point below.

  1. What is the scientific basis on which temperatures were chosen?  Reference is required.

Reply: Thank you for your reminding, the suitable temperature of S. marmoratus is constantly changing during its growth period, ranging from 13 to 26 ℃ (Xu, 1999). In addition, we also combined the results of the pre-experiment, and finally selected three representative temperatures in the paper as the experimental temperature. We have revised the expression here in the manuscript and added relevant references, please check line 318 in the revised manuscript.

  1. Line 307, oxygen was continuously charged.  Dissolved oxygen value should be provided.

Reply: Thank you for your reminding, we checked the test records from the time, and have modified the expression here and indicated the dissolved oxygen value, please check line 316 in the revised manuscript.

  1. Why did the authors choose liver tissue?  Is there any relation between it and the Hsp70 member’s expression?

Reply: Thank you for your advice, The reason we chose liver as the sample material is because liver is very important to the immunity of fish, and the liver has an important connection to the immune system. They have been found to activate immune responses, Hsp70 and HSP groups play an active role in alerting immune cells to potential threats such as the presence of cancer or invasion by pathogens (Lubkowska et al., 2021). In addition, liver has been used as a primary sampling target in many related studies of fish’s Hsp70 gene family (Xu et al., 2018).

  1. It is not mentioned the method used to euthanize the fish.  Ethics rules are mandatory.

Reply: Thank you for your advice, we added the relevant description about the method used to euthanize the fish, please check line 322-323 in the revised manuscript.

  1. Keywords should be changed to be different from the title words

Reply: Thank you for your advice, we have changed the keywords section, please check line 23 in the revised manuscript.

  1. The introduction is too short.  Need to add some effects of different temperatures on gene expression and should be ended with experiment aims.

Reply: Thank you for your reminding, we have expanded the introduction of some relevant studies on temperature and the Hsp genes, and revised the last paragraph of the introduction to end with research purposes. Please check line 47-54 in the revised manuscript.

  1. Moderate English changes are required in all manuscript

Reply: Thank you for your advice, we have asked professional native English speakers to revise the article.

Thank you again for your hard work!

References:

Xu, M., PRELIMINRY STUDY ON THE FISHRDES BIOLOGY OF SEBASTISCUS MARMORATUS. Marine Tieheries 1999, 4, 159-162.

Lubkowska A, Pluta W, StroÅ„ska A, Lalko A. Role of Heat Shock Proteins (HSP70 and HSP90) in Viral Infection. Int J Mol Sci. 2021, 22(17): 9366. https://doi: 10.3390/ijms22179366

Xu, K.; Xu, H.; Han, Z. Genome-Wide Identification of Hsp70 Genes in the Large Yellow Croaker (Larimichthys crocea) and Their Regulated Expression Under Cold and Heat Stress. Genes 2018, 9, 590. https://doi.org/10.3390/genes9120590

Reviewer 2 Report

The manuscript is written well and the results are presented correctly.

My comments are here:

Please improved the quality of Figures 2 and 5.

Figure 6 is not needed in a main document but it can be placed as a supplementary figure.

Figure 8: expression profile of liver at 20 °C must be included in figure 8.

Lines 303-311, Fish culture and their maintenance parameters are missing. How were fish confirmed as specific pathogen free and how many fish were used for this purpose? What was the water parameters? How were fish fed? Which type of feed was used and how they feed?

Line 312, please add RIN value of RNA samples. They did not use DNase to remove DNA contamination for the samples.

Lin3 314, what was total RNA concentration for RNA-seq.

Line 390, authors used only one reference gene β-actin for normalize the qRT-PCR data, however, three reference genes are needed as per the international guidelines.

The raw RNA-seq reads did not deposit to NCBI data bank. Authors must provide accession number of submitted sequences data in the manuscript.

Line 423, Animal ethics permission number is not provided in Institutional Review Board Statement.

Legends are missing in Tables, 1, 2, 3 and Figure 9.

Author Response

Dear reviewer,

Thank you for your valuable comments and for giving us the opportunity to revise the manuscript. We have carefully read your review comments and respond to your concerns point by point below.

  1. Please improved the quality of Figures 2 and 5.

Reply: Thank you for your reminding, we have re-uploaded a higher quality pictures, please refer to the revised manuscript.

  1. Figure 6 is not needed in a main document but it can be placed as a supplementary figure.

Reply: Thank you for your advice, we changed Figure 6 to supplement the figure and modified the relevant expressions in the manuscript.

  1. Figure 8: expression profile of liver at 20 °C must be included in figure 8.

Reply: Thank you for your advice, we reworked this image, and now it's Figure 7, please check line 209 in the revised manuscript.

  1. Lines 303-311, Fish culture and their maintenance parameters are missing. How were fish confirmed as specific pathogen free and how many fish were used for this purpose? What was the water parameters? How were fish fed? Which type of feed was used and how they feed?

Reply: Thank you for your advice, It was a mistake in our description. All the fish were not fed and kept on an empty stomach to reduce the interference of external factors. We explain in the article, please check line 315 in the revised manuscript.

  1. Line 312, please add RIN value of RNA samples. They did not use DNase to remove DNA contamination for the samples.

Reply: Thank you for your reminding, we have added the RIN value of RNA samples, please check line 325 in the revised manuscript.

  1. Lin3 314, what was total RNA concentration for RNA-seq.

Reply: Thank you for your reminding, we have added the total RNA concentration, please check line 326 in the revised manuscript.

  1. Line 390, authors used only one reference gene β-actin for normalize the qRT-PCR data, however, three reference genes are needed as per the international guidelines.

Reply: Thank you for your advice, Initially we used three internal reference genes for qRT-PCR, the result of β-actin was the most obvious. We looked at a number of relevant studies in the literature, most of which also used one internal reference gene.

  1. The raw RNA-seq reads did not deposit to NCBI data bank. Authors must provide accession number of submitted sequences data in the manuscript.

Reply: Thank you for your reminding, we have upload our raw RNA-seq data to National Genomics Data Center (CNCB: https://ngdc.cncb.ac.cn/), and could be accessed with BioProject No. PRJCA019415 (https://ngdc.cncb.ac.cn/gsub/submit/bioproject/PRJCA019415). We have added the description to the Data Availability Statement in the revised manuscript.

  1. Line 423, Animal ethics permission number is not provided in Institutional Review Board Statement.

Reply: Thank you for your advice, we havee added the animal ethics permission number in the section of Institutional Review Board Statement, please check line 439 in the revised manuscript.

  1. Legends are missing in Tables, 1, 2, 3 and Figure 9.

Reply: Thank you for your reminding, we have revised here in new manuscript.

Thank you again for your hard work!

Round 2

Reviewer 1 Report

·       I thank the authors for their efforts, Despite this; there are still some grammatical errors as in the abstract line 13. The sentence should be revised

·       Secondly, some of the mentioned amendments do not match the line number, please check again

needs minor corrections

Reviewer 2 Report

The authors have made all the necessary edits. The manuscript now appears much better readable.